# Virus-Mediated Inhibition of Apoptosis in the Context of EBV-Associated Diseases: Molecular Mechanisms and Therapeutic Perspectives

**DOI:** 10.3390/ijms23137265

**Published:** 2022-06-30

**Authors:** Zbigniew Wyżewski, Matylda Barbara Mielcarska, Karolina Paulina Gregorczyk-Zboroch, Anna Myszka

**Affiliations:** 1Institute of Biological Sciences, Cardinal Stefan Wyszyński University, Dewajtis 5, 01-815 Warsaw, Poland; a.myszka@uksw.edu.pl; 2Institute of Veterinary Medicine, Warsaw University of Life Sciences—SGGW, Nowoursynowska 166, 02-787 Warsaw, Poland; matylda_mielcarska@sggw.edu.pl (M.B.M.); karolina_gregorczyk_zboroch@sggw.edu.pl (K.P.G.-Z.)

**Keywords:** Epstein-Barr virus, apoptosis, BHRF1, BARF1, LMP-2A, EBNA-3C, EBER, microRNA, Burkitt’s lymphoma, nasopharyngeal carcinoma

## Abstract

Epstein-Barr virus (EBV), the representative of the *Herpesviridae* family, is a pathogen extensively distributed in the human population. One of its most characteristic features is the capability to establish latent infection in the host. The infected cells serve as a sanctuary for the dormant virus, and therefore their desensitization to apoptotic stimuli is part of the viral strategy for long-term survival. For this reason, EBV encodes a set of anti-apoptotic products. They may increase the viability of infected cells and enhance their resistance to chemotherapy, thereby contributing to the development of EBV-associated diseases, including Burkitt’s lymphoma (BL), Hodgkin’s lymphoma (HL), gastric cancer (GC), nasopharyngeal carcinoma (NPC) and several other malignancies. In this paper, we have described the molecular mechanism of anti-apoptotic actions of a set of EBV proteins. Moreover, we have reviewed the pro-survival role of non-coding viral transcripts: EBV-encoded small RNAs (EBERs) and microRNAs (miRNAs), in EBV-carrying malignant cells. The influence of EBV on the expression, activity and/or intracellular distribution of B-cell lymphoma 2 (Bcl-2) protein family members, has been presented. Finally, we have also discussed therapeutic perspectives of targeting viral anti-apoptotic products or their molecular partners.

## 1. Introduction

Apoptosis, a type I cell death, plays an essential role in the development of the organism and is necessary for its proper function. The aforementioned process normally happens during organ morphogenesis as a mechanism responsible for the controlled elimination of unwanted cells [1]. Apoptosis may be inscribed in scenario of natural growing and aging, but it also occurs in response to a wide range of incidental factors and conditions, e.g., high temperature [2,3], UV radiation [4,5,6] as well as an exposition to alcohol (ethanol) [7,8] or toxic metal ions [9,10].

Apoptosis is also the biological phenomenon that contributes to defense against intracellular infectious agents, both bacteria and viruses. Programmed cell death frustrates intracellular replication of pathogens, counteracting the development of disease. Therefore, bacteria and viruses have evolved molecular mechanisms to inhibit apoptosis and maintain the environment of their effective multiplication [11,12,13,14,15,16,17]. Research revealed anti-apoptotic activity of intracellular bacteria, *Chlamydia trachomatis*, [11] as well as of a wide range of viruses, such as vaccinia virus (VACV) [16,17], hepatitis B virus (HBV) [15,18], hepatitis C virus (HCV) [12,14], Kaposi’s sarcoma-associated herpesvirus (KSHV) [19] and Epstein–Barr virus (EBV) [13]. The last two pathogens, γ-herpesviruses, are causative agents of lytic and persistent infections that may lead to the development of several diseases including malignancies [19,20].

Research has demonstrated that the EBV genome encodes a few anti-apoptotic proteins and non-coding RNAs. The viral products contribute to the development of EBV-associated malignant diseases. In this paper, we have reviewed the molecular mechanism of EBV-mediated regulation of apoptosis in the context of virus-driven pathological processes. The therapeutic potential of some apoptosis-based treatment strategies has been also discussed.

The work has been written to systematize the current knowledge about the anti-apoptotic activity of EBV, synthesize different information spread throughout the scientific literature across multiple disciplines, and organize it into the form of an accessible review. It provides the consistent reconstruction of the molecular networks that determine the fate of EBV-infected cells. The number of facts described in our work provides the rationale for focusing on the development of anti-viral therapies that would antagonize the anti-apoptotic factors of EBV and lead to the elimination of infected cells within the human host. Thus, the article has been prepared with the intention to guide and inspire the new research on effective strategies against EBV infection.

## 2. EBV and Its Pathogenicity

Epstein–Barr virus (EBV), a member of the *Herpesviridae* family and γ-*Herpesvirinae* subfamily, is a virus widely distributed in the human population. A total of 95% of people worldwide are estimated to carry EBV. The virion contains a 170–175 kb linear dsDNA genome [21] enclosed in an icosahedral capsid of 100–120 nm in diameter. The nucleocapsid is surrounded by a tegument and encircled by a lipid envelope decorated with glycoproteins. Viral DNA, comprising approximately 90 reading frames [22], is flanked by terminal direct repeats (TRs) and divided into short and long unique sequence domains by internal repeat sequences (IRs) [23]. EBV replicates in B, T, and natural killer (NK) lymphocytes as well as in epithelial cells [21,24]. The virus is characterized by the ability to establish persistent infection. The life cycle of EBV includes primary infection followed by latency and lytic replication [21,25]. The adaptations of EBV to establish latent infection are associated with the oncogenic properties of this pathogen [26].

EBV may be carried by healthy individuals without any symptoms of its presence. However, immunosuppression accompanying acquired immune deficiency syndrome (AIDS) or transplantation treatment favors EBV-driven fatal lymphoproliferative disease [21,25]. Moreover, EBV may contribute to the development of several malignancies, such as B cell lymphomas (i.e., Burkitt’s lymphoma (BL) [27,28,29,30,31], Hodgkin’s lymphoma (HL) [32,33], and diffuse large B-cell lymphoma (DLBCL) [34,35,36]), lung carcinoma (LC) [37,38,39], gastric cancer (GC) [40,41,42], nasopharyngeal carcinoma (NPC) [43,44,45], T cell lymphoma [46], T/NK cell lymphoma (NKTL) [47,48], and AIDS-related primary central nervous system lymphoma (AR-PCNSL) [49]. EBV has also been detected in the saliva of patients with systemic lupus erythematosus (SLE), and found to be associated with one of its symptoms (i.e., oral lesions) [50]. In BL development, the viral products complement with a cellular protein named c-myelocytomatosis oncogene product (c-Myc). The disease results from the deregulation of c-Myc expression following the translocation of its gene into immunoglobulin (Ig) loci. EBV synergizes with c-Myc in pathogenic activity by preventing apoptosis of the mutant cell [51].

The virus encodes proteins that promote the progression of EBV-associated diseases. Some of them, BamH1 fragment H rightward facing (BHRF1), BamH1-A reading frame-1 (BARF1), latent membrane proteins (LMP)-1 and -2A, EBV nuclear antigen (EBNA)-1, -2, -leader protein (LP), -3A and -3C, inhibits apoptosis of infected cells. Research has also revealed the anti-apoptotic role of EBV-encoded small RNAs (EBERs) and microRNAs (miRNAs) of EBV (Figure 1). Therefore, treatment strategies discussed in medical literature include therapies aimed at inducing apoptosis in infected cells [52,53,54].

EBV infection may lead to the host cell transformation, a process associated with changes in the levels of several viral proteins: BHRF1, EBNA-1, -2, -Leader Protein (LP), -3A, -3B, 3C and LMP-1, -2A, -2B. The EBV products include also non-coding RNAs: EBERs and microRNAs (miRNAs) including BamHI A rightward transcripts (miR-BARTs) and miR-BHRF1 molecules [13]. There are a few infection patterns with different viral gene expression profiles (Figure 2). One of them, the Latency I pattern, is restricted to the expression of EBNA-1, EBERs, and miR-BARTs. The small set of EBV products implicates limited anti-apoptotic effect of EBV infection. Latency I program occurs in BL tumor cells. Another expression pattern, Latency II, is characterized by the expression of three LMPs (LMP-1, -2A, and -2B) and the products present in Latency I. The program of Latency II is realized in HL, NPC, and NKTL. The exclusive expression of BARF1 has also been detected in the NPC cells displaying this program. Next, a latent viral promoter Wp-restricted latency includes synthesis of BHRF1, EBNA-1, -LP, -3A, -3B, -3C, EBERs, and miR-BARTs. This pattern is characteristic of some BL tumor cells. Another program, Latency III, is distinguished by the expression of the full range of proteins and non-coding RNAs (BHRF1, EBNA-1, -2, -LP, -3A, -3B, 3C, LMP-1, -2A, -2B, EBERs, miR-BARTs, and miR-BHRF1s), and observed in vitro in resting B-lymphocytes converting into lymphoblastoid cell lines (LCLs). This pattern occurs also in DLBCL [13,42,55,56,57,58,59,60].

In some cases, the establishment of latency may be followed by a switch to a productive cycle aimed at the formation and release of progeny virions [61,62]. The lytic cycle consists of three stages: the immediate-early (IE) phase, the early one, and the late one. In the first phase, the IE proteins are involved in the transcription of the viral genetic material. Next, the products of the early genes act to mediate DNA replication. In the third stage of the lytic cycle, the late proteins are engaged in the formation of progeny virions. The complete productive cycle ends in the infected cell lysis [63]. The switch between the latent period and the lytic cycle depends on the expression of the genes encoding two IE products, BamHI Z EBV replication activator (ZEBRA/BZLF1) and BRLF1 [62,64]. These genes are inactive during the latent infection, whereas upon the lytic phase, they undergo activation. ZEBRA and BRLF1 play the role of transcription factors responsible for the induction of the expression of the early genes encoding a set of lytic proteins, including BHRF1, BARF1, DNA primase (BSLF1), mRNA export factor ICP27 homolog (BMLF1/BSLF2), major DNA-binding proteins (BALF)2 and 5, primase-associated factors (BBLF)2/3 and 4, DNA polymerase processivity factor (BMRF1), thymidine kinase (TK), uracil-DNA glycosylase (BKRF3) and tegument protein (BKRF4) [62,65,66,67,68]. The replication of viral DNA, occurring in the early phase of the lytic cycle, is followed by the production of the late proteins (i.e., structural molecules such as major tegument protein (BNRF1), viral capsid antigen (VCA), major capsid protein (MCP), envelope glycoprotein gp350 (BLLF1), and envelope glycoprotein B (gB)). The progeny virions are assembled from the late viral products. Interestingly, the viral particles leave the nucleus and the cell, acquiring and losing sequentially the envelopes derived from the inner nuclear membrane and Golgi apparatus, respectively. The release of the virions to the extracellular environment occurs via the fusion of the organelle-derived envelope with the plasma membrane [62,66].

The lytic cycle leads to the multiplication of the virus and its effective spreading between the host cells. As a way of the carcinogenic pathogen propagation, it may favor the EBV-associated oncogenesis, contributing to the progression of both lymphoid and epithelial malignancies. Moreover, the individual viral lytic proteins may perform various oncogenic activities. They have been found to induce the genomic aberration, promote the expression and secretion of pro-inflammatory cytokines, counteract NK cell and CD8^+^ T cell response, increase angiogenesis and tumor invasiveness, as well as down-regulate major histocompatibility complexes (MHCs) and suppress MHC class II-restricted antigen presentation [63].

BHRF1 and BARF1 are present in the infected cells in some cases of latency, but also during the lytic cycle. The anti-apoptotic activity of these proteins increases the viability of the cell and delays its death in order to preserve the environment of replication and enable the completion of the progeny virions formation [62,69]. BHRF1 and BARF1 are synthesized in the early phase of the lytic cycle. *BHRF1* gene is also expressed during the transient pre-latent abortive lytic cycle, a phase that precedes the latent period of infection [62,70].

## 3. The Anti-Apoptotic Activity of EBV Latent Proteins in the Development of Virus-Driven Diseases

### 3.1. BHRF1

BHRF1 is a homolog of cellular B-cell lymphoma 2 (Bcl-2) protein [71], a member of the Bcl-2 protein family that includes positive and negative regulators of apoptosis [72]. Host Bcl-2 protein is the anti-apoptotic one [73]. It counteracts the induction of the intrinsic apoptosis pathway by preventing permeabilization of the outer mitochondrial membrane (OMM) and consequent release of pro-apoptotic molecules, such as cytochrome c [74], apoptosis-inducing factor (AIF) [75], endonuclease G (Endo G) [76], and second mitochondria-derived activator of caspase/direct inhibitor of apoptosis-binding (SMAC/DIABLO) protein [77]. BHRF1 shares 38% amino acid sequence homology with human Bcl-2. Moreover, the viral protein is predicted to contain three Bcl-2 homology (BH) domains, BH1-BH3 [78]. The spatial structure of BHRF1 is adapted to binding and blocking pro-apoptotic members of the Bcl-2 protein family, such as Bcl-2-like protein 11 (Bim), BH3-interacting domain death agonist (Bid), p53 up-regulated modulator of apoptosis (Puma) and Bcl-2 homologous antagonist/killer (Bak). BHRF1 comprises helices α2-α5 that arrange themselves into hydrophobic surface groove responsible for binding BH3 domains of protein targets [78,79,80]. The expression of BHRF1 takes place during the viral lytic cycle, enabling the host cell to survive and supply EBV with biochemical compounds and energy to replicate. Research showed that the viral Bcl-2 homolog is also synthesized during the latent period of infection [81]. The EBV BHRF1 transcripts were identified in B cell, T cell, and NK/T cell lymphomas as well as in NPC tissue [78,82,83].

The impact of BHRF1 protein on cell vitality was confirmed by several early and recent studies. Research showed that BHRF1 is able to protect the cell by conferring resistance to various apoptotic stimuli including γ-radiation and chemical compounds such as staurosporine, ionomycin, etoposide, roscovitine, methotrexate, and anti-IgM Fab2 antibody fragments [13,84,85,86,87,88].

Kelly et al. [86] have provided evidence that BHRF1 may contribute to the B cell transformation and promote the development of lymphoid malignancies. The studies have linked strong apoptosis resistance of the EBV-infected BL cells to the Wp-restricted expression of BHRF1. In order to test the cell-protective role of BHRF1, the team used the BL cell lines retaining the Latency I program of EBV gene expression, Akata-BL, and Sav-BL. The cells were transfected with a DNA vector carrying doxycycline (dox)-regulatable expression system comprising BHRF1-coding sequence and then treated with various concentrations of dox and the apoptosis inducer, ionomycin. Transfectants of both cell lines were completely resistant to the apoptotic stimulus. The studies also revealed that exogenous BHRF1 protects Sav-BL cells from anti-IgM-induced apoptosis [86].

Fitzsimmons et al. [13] have used human BL and mice Eµ-Myc lymphoma-derived cell lines as a model of c-Myc-dependent lymphomas, a category of malignancies that includes BL. The team observed the influence of BHRF1 expression on the ability of the cell to withstand apoptotic stimulation. BHRF1 turned out to bind and inhibit Puma, Bim, Bid, and Bak. The findings have suggested that in the context of c-Myc-driven aggressive lymphomas, the cell-protective function of BHRF1 results from the synergistic effect of the down-regulation of the activity of the wide range of pro-apoptotic proteins. The team utilized the site-directed mutagenesis method to identify amino acid residues essential for the anti-apoptotic activity of BHRF1. Phenylalanine, glycine, and arginine at amino acid positions 72, 99, and 100, respectively, were found to be important for the cell-protective abilities of BHRF1 [13].

Song et al. [88] have analyzed the molecular mechanisms that contribute to the development of NPC, an epithelial cell-derived carcinoma occurring in the nasopharynx. The studies allowed to link NPC tumorigenesis to BHRF1 activity and its impact on the mitochondria of the transformed cells. BHRF1 was found to migrate to these organelles and induced mitochondrial membrane permeabilization transition (MMPT) in a way dependent on the matrix protein, cyclophilin D. BHRF1-driven MMPT, and consequent mitochondrial distortion and dysfunction intensified reactive oxygen species (ROS) production ending in the activation of mitophagy. This molecular scenario prevented apoptosis and promoted NPC tumorigenesis. As reported by the team, in NPC lines transfected with a vector carrying BHRF1-coding sequence, the apoptosis rates were substantially lower, compared to the wild-type ones. The findings suggest that EBV infection and BHRF1 expression favor the development of NPC by promoting pro-carcinoma mitophagy at the expense of apoptosis [88].

Recent research by Vilmen et al. [71] has shed light on the mechanisms responsible for the influence of BHRF1 on mitochondria. The study has revealed that BHRF1 significantly decreases the level of phosphorylation of dynamin-related protein 1 (Drp1). The unphosphorylated form of Drp1 localized to mitochondria and mediated their fission. BHRF1 has also been found to interact with beclin 1 (BECN1) and stimulate autophagy. The fragmented mitochondria clustered into mito-aggresomes, the large aggregates positioned in the juxtanuclear region of the cytoplasm. According to a model proposed by the team [71], the combination of BHRF1-dependent processes (i.e., mitochondrial fission, mito-aggresomes formation, and autophagy induction leads to the enclosure of the abovementioned organelles in the autophagosomes and finally results in the mitophagy). The engulfment of the mitochondria occurs via the PTEN-induced kinase 1 (PINK1)-parkin RBR E3 ubiquitin protein ligase (PRKN)-ubiquitine-dependent manner. The immunological implication of this molecular scenario is disruption of molecular pathway dependent on mitochondrial antiviral signaling (MAVS) protein. MAVS, a molecule located at the surface of the mitochondrion, transduces the signal to induce the synthesis of type I interferon (IFN) [71].

### 3.2. BARF1

BARF1 is another EBV product that shows homology to Bcl-2. Research on EBV has revealed the oncogenic potential of the viral protein as well as its immunomodulatory activity. In the context of latent viral infection, BARF1 expression occurs in GC and NPC cells [69]. The latter ones were shown to express ΔNp63α which is an isoform of epithelial differentiation marker p63, a transcription factor representing the p53 family. ΔNp63α has an affinity for BARF1 promoter and contributes to the establishment of EBV-driven constitutive expression of viral oncogenic protein [65]. In B-lymphocytes, BARF1 expression accompanies lytic infection but does not happen during the latent one, except in some cases of BL and NKTL documented in West Africa [69].

BARF1 molecule comprises N-terminal and C-terminal domain, each one of them approximately 100 amino acid residues in length. The first one is responsible for the malignant transformation of EBV-infected cells. Sheng et al. [89] used rodent Balb/c3T3 fibroblasts to study the mechanism of BARF1-dependent immortalization of the cell. Mutational analysis was utilized to determine the protein region required for malignant conversion, and the first 54 amino acid residues of the N-terminal domain were found to determine the oncogenic activity of BARF1. Research has revealed that the pointed region is responsible for the resistance of the cells to apoptotic stimulus (i.e., serum deprivation). In the light of the findings presented in the report, the N-terminal fragment of BARF1 induces expression of the cellular Bcl-2 protein. Taken together, the obtained data suggest a link between the malignant transformation of the EBV-infected cells and the synergistic activity of two anti-apoptotic protein homologs of viral and cellular origin, respectively [89].

Another study by Sheng et al. [90] has examined the influence of BARF1 on the level of Bcl-2 in EBV-negative Akata BL cells. BARF1 was found to stimulate Bcl-2 expression and favor tumorigenesis. After introduction into the severe combined immunodeficient (SCID) mice, BARF1-expressing cells developed into a tumor. In vitro studies have shown that BARF1 expression increases the viability of Akata BL cells and strengthens their resistance to apoptosis during 96 h incubation in a medium enriched with 0.1% fetal calf serum (FCS) [90].

Wang et al. [91] have determined the anti-apoptotic activity of BARF1 in GC cells. EBV-negative GC cell line, BGC823, was transfected with the BARF1 expression vector. BARF1 expression protected the cells from taxol-induced apoptosis. Transfectants were more resistant to apoptotic stimuli, compared to nontransfected control cells. The substantial decrease in sensitivity to taxol was the result of a considerable change in the expression of two Bcl-2 family members, anti-apoptotic Bcl-2 and pro-apoptotic Bcl-2-associated X protein (Bax). The first one was over-expressed whereas the second one underwent down-regulation. Elevated Bcl-2/Bax ratio was accompanied by a lower intracellular level of cleaved poly(ADP-ribose) polymerase (PARP) [91] which is considered to be a hallmark of apoptosis [92,93].

### 3.3. LMP-2A

LMP-2A is another EBV protein that may exert an essential influence on intracellular signal transduction and determine the fate of the infected cell. The expression of LMP-2A was observed in B cell lymphomas, BL, and HL [94,95,96]. In B-lymphocytes carrying EBV, LMP-2A mimics some downstream molecular events of the B cell receptor (BCR) signaling pathway, favoring cell survival and viral persistence [97,98,99,100,101]. Anti-apoptotic LMP-2 was also found to co-constitute EBV tumorigenic potential. In the light of scientific reports, the activity of LMP-2A synergizes with oncogenic factors and thus contributes to the malignant transformation of the cell [102,103].

The ability of viral LMP-2A to modify and partly mimic BCR signaling seems to contribute to EBV persistence in the host reservoir of long-lived resting memory B cells (MBCs). The intracellular factors involved in the BCR-controlled differentiation of B-lymphocytes, are integrated into the viral strategy for the long-term colonization of the host. MBCs serve as a refuge for EBV that hides inside them from the immune response. The viral adaptation to interplay with the host B-cells, based on the utilization of the BCR signaling network, extends the duration of the EBV infection for the entire lifetime of the host. Thus, the molecular mechanisms determining the immunocompetence of the infected organism may be hijacked by the pathogen and, paradoxically, contribute to the establishment of latent infection instead of performing an anti-viral function [102].

LMP-2A molecule comprises three domains. The first of them, the N-terminal cytoplasmic signaling domain (CSD), is able to interact with E3 ubiquitin ligases as well as with Src homology (SH)2 domain-possessing proteins including spleen tyrosine kinase (SYK) [94,104]. CSD contains an immunoreceptor tyrosine-based activation motif (ITAM), the region responsible for modulation of BCR signaling [94,105,106,107] As was shown by Lu et al. [106], in human epithelial cells, tyrosine residues within the sequence of ITAM are essential for SYK activation, and SYK-LMP-2A interaction may contribute to the development of NPC [106]. The second LMP-2A domain consists of 12 transmembrane segments (TMs) anchoring the viral protein to the cellular membrane. The C-terminus of LMP-2A defines a 27-amino acid domain that is involved in the aggregation of LMP2 isoforms to homo- and heterodimers [94,108].

In the recent studies of Fish et al. [102], LCLs and mouse lymphoma cell lines were used to characterize the mechanism of anti-apoptotic activity of LMP-2A. LCLs carrying wild-type of EBV (EBV-WT) were compared to human peripheral blood B cells infected with the mutant variant of the virus, generated via the knock-out of the LMP-2-encoding gene (LMP-2A KO). Site-specific, quantitative phosphoproteomic profiling revealed either concordant or discordant effects of LMP-2 and BCR on posttranslational modification of B cell proteins. LMP-2 stimulation counteracted, among others, tyrosine phosphorylation of protein kinase C delta (PRKCD), whereas engagement of BCR promoted this modification. On the other hand, stimulation of either LMP-2A or BCR concordantly led to tyrosine phosphorylation of a set of BCR signaling effectors including SYK, Bruton’s tyrosine kinase (BTK), and phosphatidylinositol-specific phospholipase Cγ2 (PLCγ2) [102].

Meanwhile, previous studies on B-cells showed that all of these three enzymes are involved in the inhibition of apoptosis. Tyrosine phosphorylation of SYK is the pathogenic event observed in chronic lymphocytic leukemia (CLL) B-lymphocytes. In vitro experiments by Gobessi et al. [109] have determined the mechanism of the anti-apoptotic action of SYK. The use of SYK inhibitor, R406, counteracted the BCR-dependent increase in intracellular level or activity of anti-apoptotic factors including protein kinase B (PKB)/Akt, extracellular signal-regulated kinases (ERKs), and myeloid leukemia cell differentiation 1 (Mcl-1) protein, a pro-survival member of Bcl-2 family [109]. In the light of other studies, activation of PKB has a negative impact on Bax level and promotes Bcl-2 expression in the cell [110]. Similarly, ERK promotes transcription of the *BCL2* gene and contributes to the enhancement of the pro-survival activity of the viral protein [111,112].

SYK phosphorylation leads to downstream activation of BTK and PLCγ2 [113,114]. The anti-apoptotic function of BTK, another B-cell enzyme undergoing tyrosine phosphorylation in response to stimulation of LMP-2A, has been also confirmed. Merchant and Longenecker [115] used the BTK-deficient mice to determine the role of BTK in the pro-survival effect of LMP-2A on B-cells. LMP-2A was found to increase the viability of B-lymphocytes via both BTK-dependent and BTK-independent pathways [115]. Anderson et al. [116] have shown that, in splenic B-cells, LMP-2A counterfeits BCR signaling to increase the expression of B-cell lymphoma-extra-large protein (Bcl-xL) [117], another mitoprotective factor that belongs to Bcl-2 family, antagonizes pro-apoptotic molecules such as Bax and Bak, and counteracts intrinsic pathway of apoptosis [118]. Meanwhile, early studies by Solvason et al. [119], performed on mice carrying a mutation in X-linked gene encoding BTK, have determined the anti-apoptotic role of BTK and Bcl-xL in anti-Ig stimulated splenic B-lymphocytes. B-cells lacking BTK were characterized by the inability to complete the cell cycle as well as by increased sensitivity to apoptosis. Meanwhile, the level of Bcl-xL in BTK-deficient B-lymphocytes was lower in comparison to control, non-mutated ones. In the light of the obtained results, the lack of BTK may lead to substantial impairment of B-cell capacity to respond to anti-Ig-stimulation by up-regulation of Bcl-xL. Ectopic expression of Bcl-xL in anti-Ig-stimulated mutants decreased their apoptotic potential and canceled the inhibitory effect of BTK deficiency on cell cycle progression [119]. PLCγ2, another enzyme undergoing LMP-2A-dependent phosphorylation, is also involved in molecular signaling that determines the B-cell apoptotic potential. As reported by Bell et al. [120], B-lymphocytes lacking PLCγ2 exhibit decreased viability and under-expression of Bcl-2 [120]. Taken together, the aforementioned findings suggested that LMP-2A mimics BCR signaling to maintain cell viability by promoting tyrosine phosphorylation of a set of anti-apoptotic enzyme proteins, SYK, BTK, and PLCγ2.

In addition to quantitative phosphoproteomic profiling, Fish et al. [102] carried out RNA sequencing of both EBV-WT and LMP-2A KO B-lymphocytes. The latter ones were subjected to anti-Ig-stimulation or left untreated. Analysis of transcriptomic data revealed that LMP-2A signaling promotes transcription of anti-apoptotic genes such as *BCL2L10*, and down-regulates pro-apoptotic ones including *BIM* and *BNIP3L*. RNA sequencing was consistent with mass spectrometry (MS). In the light of MS measurements, the stimulation of LMP-2A leads to a substantial increase in Bcl-xL and Bcl-2-like protein 10 (Bcl-2L10) with a remarkable decrease in Bim. Both transcriptome and proteome analysis indicated the LMP-2A-dependent change of the ratio of pro-survival and pro-apoptotic cellular factors in favor of the former [102].

Further analysis has revealed the c-Myc-dependent mechanism promoting survival and hyperproliferation of B-lymphocytes. In vitro studies have shown that LMP-2A synergizes with cellular oncogenic factors, contributing to EBV-driven malignancy. EBV-WT and LMP-2A KO LCLs were transfected with retroviral vectors encoding two green fluorescent protein (GFP)-tagged factors, c-Myc and cyclin D3 (CCND3), the latter in the oncogenic mutant variant generated by the substitution of tyrosine with alanine at position 283 (T283A). Over-expression of c-Myc and CCND3 were performed either separately or in combination. LCLs producing LMP-2A and over-expressing mutant CCND3 exhibited increased proliferative potential, whereas LMP-2A KO B-cells were characterized by a normal growth despite transfection with the oncogenic variant of CCND3. LMP-2A LCLs transfected with c-Myc-encoding plasmid were not able to survive 9 days of incubation in co-culture with non-transduced parental LCLs, whereas EBV-WT c-Myc-overexpressing LCLs exhibited the capacity to persist under these circumstances. The most intensive proliferation was observed in EBV-WT LCLs transfected with both c-Myc and mutant CCND3. The results of the aforementioned experiment suggested that viral LMP-2A may synergize with c-Myc and mutant CCND3 to promote B-cell survival and growth, probably stimulating the expression of anti-apoptotic members of the Bcl-2 family, and antagonizing retinoblastoma protein (Rb)1 [102], a tumor suppressor involved in cell cycle regulation and responsible for the prevention of B-cells hyperproliferation [121]. The possible cooperation of mutant CCND3 with viral LMP-2A may contribute to the development of EBV-driven malignancies such as BL. This hypothesis is supported by the analyses that detected oncogenic CCND3 mutations in sporadic BL cases [122,123].

### 3.4. EBNA-3C

EBNA-3C is another EBV protein that exhibits anti-apoptotic activity. The expression of EBNA-3C was observed in immunoblastic lymphomas [124] and found to be necessary for in vitro transformation of primary B lymphocytes into LCLs [125].

EBNA-3C has the ability to interfere with intracellular p53-mediated signaling [126]. p53, the transcription factor responsible for tumor suppression, is widely described in the literature both as a cell cycle regulator and as a potent pro-apoptotic agent [127,128,129,130,131,132]. p53 may promote the expression of the pro-apoptotic members of Bcl-2 family [130] The intracellular signaling network dependent on p53 includes Puma [133], Bim [134], Bid [135], Bax [136,137], and Bak [138,139]. In approximately 50% of malignancies, p53 occurs in mutant form, whereas in the rest, it remains non-mutated and therefore capable to promote apoptosis in response to DNA damage [26].

As reported by Yi et al. [140], EBNA-3C exhibits the ability to interfere with the transcriptional activity of p53 in EBV-infected LCLs. The team observed direct interaction between these two proteins and identified the p53-binding regions within the EBNA-3C molecule. Binding assays and the immunoprecipitation technique were used to perform in vitro and in vivo experiments, respectively, and the results given by both of these methods concordantly indicated that EBNA-3C forms complex with p53. Further analyses revealed that EBNA-3C binds to p53 via N-terminal fragment comprising 61 amino acid residues at amino acid positions 130–190 and characterized by the presence of leucine zipper motif (LZM) [140]. The aforementioned protein region was previously found to be engaged in interactions of EBNA-3C with other cellular proteins including c-Myc and cyclins A, E, and D1 [26,141]. Next, the team of Yi et al. [140] used transfectants derived from human osteoblast-like cells SAOS-2 to determine the antagonistic relationship between EBNA-3C and p53 in terms of the cell viability regulation. The cells transfected with vectors encoding both EBNA-3C and p53 exhibit decreased level of apoptosis, compared to the ones carrying constructs that expressed only p53 [140].

In addition to interaction with p53, EBNA-3C is competent to interplay with the E2F1 transcription factor, another cellular protein determining the cell’s fate [26]. In the light of literature, genotoxic stress may cause the molecular machinery of the cell to induce apoptosis via the E2F1-mediated signaling pathway. E2F1 is able to counteract apoptosis and tumorigenesis, either individually or collectively through collaboration with other molecules including p53 [142,143]. Studies by Saha et al. [144] on EBNA-3C-deficient LCLs have determined the pivotal role of EBNA-3C in preventing DNA damage-induced cell death. LCLs lacking EBNA-3C exhibited increased sensitivity to genotoxic stress, a 12 h exposure to 5 µM etoposide, compared to the control cells expressing EBNA-3C. The silencing of EBNA-3C expression in LCLs resulted in a significantly elevated level of apoptosis. In another experiment, the use of p53-deficient SAOS-2 cell lines transfected with plasmid vectors expressing EBNA-3C and/or E2F1 showed that the viral protein prevents apoptosis by acting independently of p53. Next, the team recognized the DNA-binding domain localized at the N-terminus of E2F1 (amino acid residues 1-243) as the site of interaction with EBNA-3C via its two distinct fragments consisting of amino acid residues at positions 100–200 and 621–700, respectively [144]. The abovementioned results were consistent with the previous reports linking E2F1-dependent apoptosis with the DNA-binding activity of E2F1 [145,146]. Indeed, Saha et al. [144] have found that EBNA-3C neutralizes E2F1 by preventing it from binding to the promoter sequence of the pro-apoptotic genes encoding p73 and apoptotic protease activating factor 1 (Apaf-1). Moreover, the viral product promoted proteolysis of E2F1 via the ubiquitin-proteasome pathway (UPP) [144]. Taken together, these findings underline the importance of EBNA-3C in B-cells immortalization, an event that contributes to BL pathogenesis [147]. The findings stated above are consistent with studies by Molina-Privado et al. [148] that have delivered observation of increased E2F1 expression in BL cell lines and specimens [148].

Another cellular target for EBNA-3C is Bim [26,29,149,150]. As the pro-apoptotic member of the Bcl-2 family, Bim is involved in c-Myc-dependent molecular signaling that decreases the viability and the oncogenic potential of the cell. In the light of scientific reports, Bim counteracts the development of B-cell malignancies [151,152]. The experiment of Egle et al. [151] on c-Myc-expressing mouse mutants has confirmed the anti-oncogenic properties of Bim in the context of B-cell leukemia pathogenesis. The transgenic mice lacking a single *BIM* allele exhibited intensified progression of the disease, compared to the non-mutated animals. These results are in line with clinical studies. Research by Richter-Larrea et al. [152], performed on Spanish patients suffering BL, has linked the epigenetic repression of *BIM* gene transcription with the decrease in complete remission rate and overall survival, compared to Bim expressing individuals. These results suggest that the pro-apoptotic activity of Bim might sensitize the tumor to chemotherapeutic treatment [152]. Meanwhile, Anderton et al. [153] have revealed antagonistic activity of EBNA-3C towards Bim. In experiments of this team, EBV-negative BL cells were treated with apoptosis inducers, nocodazole, cisplatin, or roscovitine, after the previous infection with EBV mutants carrying independent deletions of the genes encoded one of the following viral products: EBNA-2, EBNA-3A, EBNA-3B or EBNA-3C. As a result of exposure to nocodazole, the BL cells lacking EBNA-3A or EBNA-3C displayed drastically elevated levels of apoptosis, compared to the cells infected with EBV-WT, EBNA-2-knockout (KO) EBV or EBNA3B-KO EBV. Moreover, BL cells lacking EBNA-3A or EBNA-3C exhibited increased susceptibility to cisplatin- and roscovitine-induced apoptosis. These findings suggested the effective molecular collaboration of EBNA-3A and EBNA-3C in maintaining the viability of B cells under cytotoxic stresses. Further experiments showed that these two viral products synergize to decrease the pro-apoptotic member of the Bcl-2 family, Bim, by down-regulation of intracellular level of Bim-encoding mRNA [153]. Next, Paschos et al. [154] determined the mechanism responsible for the repression of *BIM* gene transcription. The treatment of EBV-negative and EBV-positive BL cells with inhibitors of histone deacetylase (HDAC) and DNA methyltransferase (DNMT), trichostatin A (TSA), and 5′-azacytidine (AZA), respectively, suggested that Bim-encoding mRNA level may be regulated via EBV-driven epigenetic modifications of the host cell chromatin. EBV was found to affect the trimethylation and acetylation of the histones associated with the *BIM* gene promoter. The virus increased the level of trimethylation of the lysine residue at amino acid position 27 of histone H3 while counteracting acetylation of histones H3 and H4. Moreover, EBV-positive B cells displayed specific methylation of CpG dinucleotides within the 5′-end CpG island (CGI) of the *BIM* gene. In addition, analysis of DNA derived from African BL biopsies indicated EBV-dependent methylation of *BIM* promoter [154]. As EBNA-3C is known to be involved in the remodeling of chromatin, it is highly possible that this viral product affects the level of Bim by changing the patterns of epigenetic modifications, especially the methylation of CGI [26].

Jha et al. [155] have determined the impact of EBNA-3C on the expression of aurora kinase B (AURK-B) [155], an enzyme that plays a role in cell division, affects cell viability, and is involved in the pathogenesis of leukemia [156]. EBNA-3C was found to stimulate the transcription of AURK-B and decrease the level of its ubiquitin-dependent proteolysis. HEK-293T cells transfected with EBNA-3C and AURK-B together were more resistant to apoptosis, compared to the ones that expressed EBNA-3C or AURK-B individually. According to a model proposed by authors [155], the cooperation of EBNA-3C and AURKB increased the viability and proliferation potential of the cells. AURKB activity, supported by EBNA-3C, prevents activation of caspases-9 and -3, counteracting cell death via mitochondria-dependent apoptosis pathway. In addition, EBNA-3C and AURKB synergize to phosphorylate the key regulator of the cell cycle, anti-tumor protein Rb, and promote its consequent degradation. EBNA-3C-AURKB collaboration seems to contribute to cell survival, growth, and EBV-driven oncogenic transformation [155].

Banerjee et al. [157] have described the interaction of EBNA-3C with an oncogenic enzyme, serine/threonine kinase Pim-1. Experiments have revealed that EBNA-3C favors transcription and translation of the *PIM-1* gene, directly interacts with the Pim-1 molecule, affects its intracellular distribution, and prevents degradation via the ubiquitin-proteasome pathway. Meanwhile, the short hairpin RNA (shRNA)-mediated silencing of the *PIM-1* gene strongly sensitized LCLs to the intrinsic apoptosis pathway. These results link the anti-apoptotic activity of EBNA3C to Pim-1 signaling that may contribute to the survival of malignant cells [157].

Another research devoted to EBV has revealed the link of the EBNA-3C activity with autophagy, an intracellular process counteracting apoptosis and promoting B cell viability. Bhattacharjee et al. [158] have compared the markers of apoptosis and autophagy in LCLs nonexpressing EBNA-3C and BJAB cells exhibiting the ability to synthesize this protein. LCLs displayed a higher level of PARP cleavage, an event that is a hallmark of apoptosis, in comparison to BJAB cells. The latter cells exhibited, however, more intense conversion of microtubule-associated protein 1 light chain 3 (MAP1LC3)-II and lower accumulation of p62, suggesting the presence of EBNA-3C-dependent autophagy. Further analyses have revealed that in resting B-lymphocytes infected with EBV and subjected to starvation, EBNA-3C promotes methylation of three lysine amino acid residues (at positions 4, 8, 27) within histone H3 to affect the expression of a set of autophagy-associated genes at the level of transcription. According to a model proposed by Bhattacharjee et al. [158], EBNA-3C favors the expression of autophagy-related genes (*ATG*s), *ATG3*, *ATG5*, *ATG7*, and *ATG16*, whose products are involved in the biogenesis of autophagosomes. Under growth factor deficiency conditions, EBNA-3C may also stimulate the expression of the DNA damage-regulated autophagy modulator 1 (DRAM1) gene [158] that encodes a stress-inducible lysosomal membrane protein responsible for the regulation of both autophagy and apoptosis [159,160]. Moreover, EBNA-3C is able to promote autophagy by transactivation of the anti-oncogenic genes encoding cyclin-dependent kinase inhibitors (CDKNs), CDKN1B, and CDKN2A, as well as death-associated protein kinase 1 (DAPK1) [158]. As previously described by Harrison et al. [161], in the reaction to the cell starvation, DAPK1 forms a complex with microtubule-associated protein (MAP)1B, and DAPK-1-MAP1B interaction leads to the induction of autophagy [161]. In addition, a recent report by Nowosad and Besson [162] has elucidated the mechanisms of CDKN1B-dependent autophagy, linking it to inhibition of the mammalian target of rapamycin complex 1 (mTORC1) kinase activity and subsequent migration of transcription factor EB (TFEB) to the cell nucleus [162].

### 3.5. Other EBV Latent Proteins: LMP-1, EBNA-1, -2, -3A and LP

The set of EBV proteins capable of promoting the survival of infected cells and contributing to virus-driven diseases is extensive. Apart from BHRF1, BARF1, LMP-2A and EBNA-3C, it includes LMP-1, EBNA-1, -2, -3A and -LP. LMP-1 and EBNA-1 have been shown to perform both anti-apoptotic and pro-apoptotic activities [126]. This paper focuses on the latter one.

LMP-1 increases cell viability by up-regulation of anti-apoptotic protein: Bcl-2, Mcl-1, Bcl-2-related protein A1 (Bcl-2A1), and tumor necrosis factor (TNF) alpha-induced protein 3 (TNFAIP3)/A20. LMP-1-mediated stimulation of expression of Bcl-2 and/or Mcl-1 seems to be one of the mechanisms favoring the survival of the BL cells [126,163,164]. LMP1 is able to interfere with TNF signaling. The viral protein localizes in the cell membrane and recruits TNF receptor (TNFR)-associated factors (TRAFs) and TNFR-associated death domain proteins (TRADDs). Interaction of LMP1 with these molecules leads to the expression of *BCL2* and *A20* genes as a consequence of activation of three signaling cascades (i.e., c-Jun N-terminal kinase (JNK), nuclear factor ĸB (NF-ĸB), and phosphoinositide 3-kinase (PI3K)/PKB/Akt pathways) [165,166]. Early clinical research by Camilleri-Broët et al. has revealed a correlated over-expression of LMP-1 and Bcl-2 in AR-PCNSL. These results suggest possible cooperation between LMP-1 and Bcl-2 in terms of promoting the host cell survival and the development of the disease [49].

EBNA-1 is the viral product that plays a role in HG, GC, and NPC development [165,167]. Moreover, it performs its anti-apoptotic and oncogenic activity in BL cells [126]. EBNA-1 is supposed to promote the proliferation of BL lymphocytes in vivo, even alone, in the context of the Latency I program, under which it is expressed in the absence of other latent viral proteins [165]. Research has shown [168] that EBNA-1 is able to interact with herpesvirus-associated ubiquitin-specific protease (HAUSP)/USP7, an enzyme that deubiquitinates and stabilizes p53, promoting p53-mediated apoptotic events. By interfering with the effect of HAUSP/USP7 on p53, EBNA-1 promotes p53 degradation and counteracts apoptotic events [168]. In vivo experiments have revealed that EBNA-1 decreases the p53 level in GC, enhances apoptosis resistance of the cells, favors their survival and outgrowth, and thus promotes the pathogenesis of the malignancy [165]. In EBV-positive BL cells, EBNA-1 cooperates with Sp1 or Sp1-like proteins to enhance the expression of anti-apoptotic protein, survivin [169] that is known to block the activity of executioner caspases-3 and -7 and thus protect the cells from death via both extrinsic and intrinsic apoptosis pathway [170]. Another mechanism of the EBNA-1-mediated anti-apoptotic effect is to bind tumor suppressor NM23-H1, a protein that stimulates transcription of the genes encoding p53 and pro-apoptotic caspases-9 and -3. EBNA-1 promotes NM23-H1 sequestration and inhibits its activity, thereby counteracting NM23-H1-dependent pro-apoptotic events [171].

EBNA-2 was found to stimulate the expression of anti-apoptotic Bcl-2A1 in LCLs. Research has shown that EBNA-2 up-regulates Bcl-2A1 in co-operation with recombination signal binding protein for immunoglobulin kappa J region (RBPJ) that is able to bind *BCL2A1* gene promoter. Transactivation of the abovementioned gene is the mechanism that seems to contribute to the development of B-cell lymphomas by promoting the survival of malignant cells [172]. Intriguingly, the presence of EBNA-2 is also associated with the increase in a set of anti-apoptotic members of the Bcl-2 family: Bcl-2, Bcl-xL, and Mcl-1 in B-lymphocytes and BL cells [126,173]. In addition to its positive effect on the expression of pro-survival molecules, EBNA-2 was found to be involved in the down-regulation of pro-apoptotic protein, Bcl-2 interacting killer (Bik), in EBV-negative BL cells [174]. Moreover, two recent reports [175,176] describe the anti-apoptotic effect of interaction between EBNA-2 and the nuclear receptor 4A1 (NR4A1) [175,176]. EBNA-2 protects the cells from death caused by etoposide and 5-fluorouracil, the chemical compounds that induce apoptosis in the NR4A1-dependent manner. In the experiments of Lee et al. [176], the LCLs transfected with lentiviral vector encoded mutant EBNA2 lacking conserved region (CR)4 were sensitive to inducers of NR4A1-mediated apoptosis, whereas the cells expressing wild-type EBNA displayed resistance to these chemicals [176].

As mentioned above, EBNA-3A cooperates with EBNA-3C to repress BIM gene transcription [177]. In addition, EBNA3A mediates the translocation of Mcl-1 to mitochondria, and thus counteracts the initiation of the intrinsic apoptosis pathway [149].

EBNA-LP is another EBV protein that participates in a complex signaling network responsible for the transformation of B-lymphocytes [178]. Scientific reports [179,180] link anti-apoptotic activity of EBNA-LP to its co-operation with HS1-associated protein X-1 (HAX-1) [179,180]. Matsuda et al. [179] has documented the formation of complexes between EBNA-LP, HAX-1, and Bcl-2 in monkey kidney epithelial COS-7 cells. These results indicate that EBNA-LP may use HAX-1 to exert an indirect impact on Bcl-2 activity, thereby decreasing the apoptotic potential of the cell [179].

## 4. The Anti-Apoptotic Activity of EBV RNA Molecules in the Development of Virus-Driven Diseases

### 4.1. EBERs

EBV EBERs, EBNA1, and EBNA2, the products of RNA polymerase III [181], are non-coding RNA molecules involved in the signaling network that determines the latency of the viral infection. They are constantly synthesized in persistently infected cells and co-constitute viral transcriptome in EBV-driven diseases such as BL, HL, NPC, and T/NK cell lymphoma [13,55,56,182,183]. EBERs are localized mainly in the nucleus, however, in the light of the studies of Ahmed et al. [184] on B-lymphocytes, they may be also transported to the cytoplasm and released from the cell via the exosome pathway [184].

EBER1 and EBER2 are polymers built of 166 and 172 nucleotides, respectively. The EBER molecules display a high level of internal complementarity and thus they are arranged into partially double-stranded structures that contain the short stable stem-loops. The spatial organization of EBERs enables them to interact with some cellular proteins, including dsRNA-dependent protein kinase (PKR). One of the EBER1 secondary structure elements, the stem-loop IV, binds to PKR to form an RNA-protein complex [184,185].

In the light of scientific reports, the anti-apoptotic activity of EBERs promotes EBV-associated malignancies such as BL [186] and NPC [187]. EBERs were found to contribute to the development of EBV-driven diseases by several mechanisms [182], including the up-regulation of Bcl-2 and consequent desensitization of the cells to apoptotic stimuli [126,186,187]. Early studies link the cell-protective role of EBERs with their suppressive effect on PKR [187,188,189,190,191]. This protein is an anti-viral, pro-apoptotic enzyme induced via the IFN-dependent manner and activated through the interaction with dsRNA molecules. PKR signaling leads to molecular events specific to both extrinsic and intrinsic apoptosis pathways, including activation of caspases-8 and -9 [192,193]. Wong et al. [187] transfected immortalized nasopharyngeal epithelial cells, NP69, with the plasmid encoding EBERs, and then transfectants were subjected to apoptotic stimulus, polyinosinic:polycytidylic acid (poly(I:C)) that is able to activate PKR and to induce PKR-dependent apoptosis pathway. The cells that expressed EBER, displayed high resistance to apoptosis, as opposed to the control, EBER-non-expressing cells. EBER-positive cells exhibited a substantial decrease in the level phosphorylation of PKR and two proteins involved in downstream signaling pathways of this enzyme: mitogen-activated protein kinase (p38 MAPK) and JNK. Finally, the presence of EBERs implicated a significant increase in the level of anti-apoptotic Bcl-2 [187]. These results are consistent with the early studies of Nanbo et al. [190] performed with the use of BL cell lines. EBV-negative EBER-expressing transfectants were found to be resistant to IFN-α-induced apoptosis since EBERs bound PKR to prevent its phosphorylation and activation [190]. Surprisingly, the abovementioned results have been questioned by Ruf et al. [181] that used BL cells to determine whether EBERs indeed are able to inactivate PKR. Experiments have confirmed that EBER-expressing BL cells confer resistance to IFN-α-induced apoptosis. However, intriguingly, they have negated the molecular mechanism that had been previously postulated to explain the anti-apoptotic effect of EBERs. The tests using phospho-specific antibodies have shown that EBERs exert no influence on the phosphorylation state of either PKR and its downstream signal molecule, the eukaryotic initiation factor-2α (eIF2α) [181]. Further investigations are required to resolve inconsistencies between the scientific reports presented above.

### 4.2. miRNAs

miRNAs are cellular or viral, single-stranded non-coding ribooligonucleotides that regulate gene expression by affecting the stability and functionality of mRNA transcripts. The miRNA sequence displays complementarity to the 3′-untranslated region of mRNA, and therefore it is able to bind it specifically. miRNAs may promote degradation of the target mRNA molecules and/or silence protein synthesis. miRNA-mediated regulation of gene expression can change the cell phenotype to promote or counteract the development of various human pathologies [194,195]. miRNA-associated diseases affect different tissues, organs, and systems of the human body, and include multiple sclerosis (MS), systemic lupus erythematosus (SLE), type II diabetes, and several malignancies such as breast, lung, gastric, and liver cancers. Moreover, miRNA is associated with infections caused by viruses, including human immunodeficiency virus (HIV), HCV, and EBV [196,197,198]. The number of miRNAs encoded by EBV exceeds 40. They have been studied and described in the context of EBV-associated diseases [58,199,200].

EVB encodes 25 miRNA precursors (pre-miRNAs) that undergo transformation into mature forms through further processing. A set of EBV miRNA transcripts can be divided into two subsets, miR-BARTs and miR-BHRF1s. These two groups of miRNAs are encoded in two different regions of the EBV genome, BART and BHRF1, respectively [58].

#### 4.2.1. miR-BARTs

miBARTs are associated with various EBV-driven malignancies including BL, HL, GC, NPC, and NKTL [13,55,56,57,58,201], and their pathogenic role is considered to be especially important in the development of epithelial malignancies [126].

Research by Marquitz et al. [202] has shown the collective influence of miR-BARTs on the expression of pro-apoptotic representative of the Bcl-2 family, Bim protein. The team transfected GC cells with two vectors, each representing a distinct fragment of the BARF region and encoding a different cluster of transcripts. miR-BART clusters, either combined or expressed individually, allowed the transfectants to survive the exhibition to apoptosis inducer, etoposide. Further analyses allowed the authors to predict putative biding sites for various miR-BARTs within the 3′ untranslated region (3′ UTR) of Bim as well as to confirm the negative effect of both clusters, especially the first one, on *BIM* gene transcription and translation. The team has also determined the contribution of individual miR-BARTs and indicated the three molecules (miR-BART-9, -11, and -12) as the stronger suppressor of *BIM* gene expression [202].

In GC and NPC cells, EBV infection manifests with a high intracellular level of miR-BART5. Meanwhile, this non-coding transcript was found to silence the expression of the pro-apoptotic member of the Bcl-2 family, Puma, and thus prevent Puma-mediated p-53-independent apoptosis. Choy et al. [196] have measured the level of Puma in EBV-positive and EBV-negative NPC cell lines and found that the presence of EBV implicates a substantial decrease in the expression of *PUMA* gene. Further analyses have proved that the suppressive effect of miR-BART on Puma is mediated by mi-BART5. Lipid-based transfection of Puma-expressing HeLa cells with the pre-miR-BART5 resulted in a substantial reduction of the level of this protein. Meanwhile, EBV-positive NLP cells displayed increased expression of Puma after the introduction of anti-miR-BART5, an oligonucleotide inhibitor of miR-BART5. Moreover, HEK293 cells carrying mi-BART5-expressing vector exhibited a low level of Puma-encoding mRNA. These results have suggested a pro-survival mechanism that may contribute to the pathogenesis of EBV-associated malignancies arising from epithelial cells [196].

Another EBV transcript, miR-BART4-5p has been reported to down-regulate the expression of Bid in GC cells. Shinozaki-Ushiku et al. [203] used a luciferase reported assay to demonstrate that miR-BART4-5p molecule is able to bind specifically the 3′ UTR of Bid mRNA. In further experiments, the EBV-negative GC cells transfected with miR-BART4-5p mimic exhibited lower expression of Bid and increased apoptotic rate in starvation conditions, compared to control. The down-regulation of Bax was performed at the level of protein translation since the loss of Bid protein was not accompanied by the decrease in Bid mRNA level. Moreover, in EBV-positive GC cells, the EBV-mediated suppression of the Bid expression was partially reversed in the presence of miR-BART4-5p inhibitor. These results gain in importance when confronted with in vivo findings that have also been presented by the abovementioned team [203]. Among a set of tissue specimens derived from GC patients, those associated with EBV infection display decreased level of apoptosis. Taken together, in vitro and in vivo experiments have suggested that miR-BART4-5p participates in GC development, affecting the expression of Bid and modifying the apoptotic potential of malignant cells [203].

Kim et al. [204] have revealed that miR-BART20-5p targets mRNA encoding a pro-apoptotic protein, Bcl-2-associated agonist of cell death (Bad). GC cells transfected with the aforementioned transcript showed decreased Bad level, and the use of miR-BART20-5p restored the normal expression of this protein. miR-BART20-5p-mediated down-regulation of Bad was found to be responsible for the cell growth promotion, apoptosis inhibition, and desensitization of the GC cells to the chemical inducers of the cell death, 5-fluorouracil (5-FU) and docetaxel [204].

Recent studies by Min et al. [199] have linked the anti-apoptotic and oncogenic role of another non-coding viral transcript, miR-BART1-3p, to the under-expression of the tumor suppressor named disabled homolog 2 (DAB2). miR-BART1-3p was found to bind to the 3′ UTR of DAB2 mRNA. Both GC cells transfected with miR-BART1-3p exhibited decreased DAB2 levels and lower apoptosis rate, compared to the control. The DAB2 knock-down also protected the cells from apoptosis. Substantial depletion of pro-apoptotic Bax in miR-BART1-3p-expressing GC cells suggested that the disruption of DAB2 expression may counteract the intrinsic apoptosis pathway. Molecular mechanisms linking the antagonism of miR-BART1-3p towards DAB2 to apoptosis signaling remain unknown [199].

Interestingly, EBV miRNA may also exert an influence on intracellular localization of Bax through targeting mRNA of its mitochondrial receptor, a key component of the translocase of the outer membrane of mitochondria (TOM) [56]. This protein, named mitochondrial import receptor subunit TOM22 homolog (TOMM22), has been reported to recruit Bax to mitochondria, and thus to be co-responsible for Bax-dependent permeabilization of OMM and consequent activation of the intrinsic apoptosis pathway [205]. Meanwhile, Dölken et al. [206] have utilized luciferase assay to confirm the affinity of miR-BART16 to 3′ UTR of TOMM22 [206]. Taken together, these findings suggest that EBV miR-BARTs play a mitoprotective role by causing both the decrease in the level of Bax and its dislocation from OMM [56,199].

In addition to the role in silencing the expression or reducing the effectiveness of pro-apoptotic Bcl-2 family proteins, miR-BARTs may also target their downstream effector molecule, caspase-3. Harold et al. [207] have reported a set of 9 miR-BART molecules to bind to 3′ UTR of caspase-3. Luciferase assay has shown that miR-BART22 exerts the largest silencing effect on caspase-3 3′ UTR reporter construct. The other effective miRNAs were miR-BART1, 2, 3, 4, 7, 8, and 13. Western blot analysis of caspase-3 level in HEK-293T cells transfected with individual miR-BARTs has confirmed the silencing potential of miR-BART22 and, to a lesser extent, miR-BART1, 2, 7, and 8 [207].

#### 4.2.2. miR-BHRF1s

Apart from miR-BARTs, under the Latency III program, EBV synthesizes three miR-BHRF1s (miR-BHRF1-1, -2, and -3) [58]. The report of Bernhardt et al. [84] has been devoted to miR-BHRF1s and their impact on the expression of BHRF1 protein. The team has determined the influence of the deficiency of BHRF1 protein and miR-BHRF1s on apoptosis degree and transformation capacity of the EBV-infected cells. The team used the virus mutant carrying triple knockout of miR-BHRF1s (Δ123 EBV) to infect B-lymphocytes. Δ123 EBV-positive cells were more susceptible to apoptosis and had a two to three fold lower mitotic rate, in comparison to the EBV-WT ones (control). The difference in cell growth and vitality was transient as it took place between 5 and 20 days of infection. Further investigation revealed that miR-BHRF1s are responsible for the regulation of the BHRF1 expression at the level of transcription and translation. Two miR-BHRF1s, miR-BHRF1-3 and to a lesser extent miR-BHRF1-2, turned out to increase the quantity of BHRF1 transcripts and protein at the early stage of infection, and thus diminish the rate of apoptosis. miRNAs possibly enhance the Wp-regulatable *BHFR1* gene transcription. The knockout of miR-BHRF1s subverted this effect. Interestingly, the development of viral infection, the transformation of B-cell, and the establishment of LCLs inversed the correlation between the intracellular level of miR-BHRF1-2 and BHRF1 protein. After 2 months after infection, Bernhardt et al. [84] observed the probable link between the processing of miR-BHRF1-2 and the following degradation of large poly-adenylated mRNA that comprised the BHRF1-coding sequence and seemed to be the source of BHRF1. In contrast to the early-infected B-cells, LCLs generated with Δ123 EBV mutant withstood apoptotic stimuli more effectively than the cells carrying EBV-WT. In the latter ones, a substantial increase and subsequent decrease in BHRF1 at the early and late stage of infection, respectively, seems to be EBV evolutionary strategy aimed to protect the cell reservoir for efficient viral replication without inducing BHRF1-targeting T cell immune response. The findings underline the importance of miR-BHRF1s for the viability of the EBV-infected B-cells and established LCLs [84].

## 5. EBV-Encoded Anti-Apoptotic Molecules as the Potential Targets for Anti-Viral Drugs: Therapeutic Perspectives

In order to establish persistent infection, EBV orchestrates molecular interaction networks inside the infected cells. The viral products are able to interfere with the host signaling pathways and redirect them towards executing the pathogen survival strategies instead of neutralizing the “intruder”. The molecules presented above exhibit anti-apoptotic activities and contribute to maintaining the viability of infected cells. The pro-survival role of EBV products is discussed in the context of EBV-associated malignancies derived from immune and epithelial cells, and thus these anti-apoptotic agents are considered potential target molecules in the treatment of the EBV-driven diseases. So far, therapies for lymphomas and other cancers have focused on inhibition of cellular regulatory proteins (i.e., the mitoprotective members of the Bcl-2 family such as Bcl-2, Bcl-xL, and Mcl-1). Clinical research has been devoted to the activity and efficacy of BH3-mimetic drugs, such as AMG 176, navitoclax, and venetoclax [208,209,210,211]. In the case of EBV-associated diseases, viral anti-apoptotic proteins seem to be the more promising therapeutic target that the cellular ones, for the sake of selectivity of treatment to infected cells [13]. Therefore, many reports suggest focusing on possible therapeutic implications of EBV latent proteins [13,53,158,212,213], however, the studies on their inhibitors are far from clinical trials and include mainly in vitro experiments [53,212,213].

Preliminary research tentatively suggests the possibility of killing the malignant EBV-infected cells by targeting viral anti-apoptotic proteins. Sun et al. [53] have shown that 17-DMAG, a representative of heat shock protein (hsp)90 inhibitors, may down-regulate the EBNA1 in LCLs and consequently sensitize the cells to apoptosis [53].

An interesting example of a treatment strategy that antagonizes the anti-apoptotic activity of EBV protein is the combined therapy using suberoylanilide hydroxamic acid (SAHA) and bortezomib. As reported by Hui et al. [52], these two drugs synergistically cause the death of LCLs and BL cells, as well as inhibit the growth of BL xenografts in mice. The authors have suggested that the therapeutic effect of combined therapy results from mechanisms counteracting EBNA-3C activity. However, SAHA and bortezomib do not target EBNA-3C directly but do affect its cellular partners. These drugs inhibit HDAC enzymes [52] that are responsible for epigenetic modification of chromatin. As mentioned before, EBNA-3C may use HDAC to affect the acetylation states of histones, and thus regulate the expression of the host genes including *BIM* [52,154]. Moreover, SAHA and bortezomib suppress the activity of Pim-1 [52].

In addition to EBV proteins, the viral miRNAs are discussed as therapeutic targets in the context of EBV-associated disease treatment [214,215]. This issue fit squarely into the general discourse on the neutralization of pathogenic miRNAs in the effective treatment of various pathologies, including both infectious and non-infectious ones. In the light of extensive studies and reports, anti-miRNA antisense inhibitors (AMOs) open up promising perspectives for medicine [195,216,217]. The AMOs’ effectiveness in counteracting anti-apoptotic, oncogenic, and pro-viral effects of EBV miRNAs, miRNA-BARTs, and miRNA-BHRF1, seems to be an issue worthy of consideration and comprehensive investigation.

In the light of the findings of Bernhardt et al. [84] that have been mentioned above, miR-BHRF1 miRNAs seem to be strong candidates for the novel target structures in BL therapies. So far, Ma et al. [54] have suggested that the therapy directed to miR-BHRF1-2 might be an effective treatment for aggressive malignancies such as DLBCL and HL. According to the authors, the inactivation of miR-BHRF1-2 could nullify the down-regulation of PR domain zinc finger protein 1/B lymphocyte-induced maturation protein-1 (PRDM1/Blimp1), a pro-apoptotic agent known to suppress the lymphomas [54].

## 6. Conclusions

EBV, a member of the *Herpesviridae* family, is able to establish latent infection for the entire lifetime of the colonized organism, and thus evade the anti-viral immune response of the host. The viral adaptation to survive within the host body includes the strategies for keeping infected cells alive, as they play a role of refuge for the dormant virus. Therefore, EBV encodes a set of anti-apoptotic products, including a group of the latent proteins (BHRF1, BARF1, LMP-1, -2A, EBNA-1, -2, 3A, -3C, -LP) and RNA molecules (EBERs, miR-BARTs, miR-BHRF1s). The pro-survival activity of these agents affects the transcription, translation, stability, and/or activity of various cellular regulatory proteins, especially the members of Bcl-2 that determine the cell potential to undergo mitochondria-dependent apoptosis. The latent proteins and non-coding transcripts of the virus can up-regulate the level and/or activity of the anti-apoptotic representatives of this family (i.e., Bcl-2, Mcl-1, Bcl-2A1, Bcl-2L10) and down-regulate the pro-apoptotic ones (i.e., Bax, Bak, Bad, Bid, Bim, Puma). EBV products affect Bcl-2 family members through several mechanisms, including direct protein-protein interactions, miRNA-mediated silencing of translation, epigenetic modifications of chromatin (histones and gene promoters), and complex interplay with various signal transduction pathways, including those dependent on p53, PKB/Akt, ERK, PI3K, JNK, PKR, SYK, and E2F1 proteins. In addition, EBV stimulates the expression of the survivin that protects the cells from death via both extrinsic and intrinsic apoptosis pathways, since it inhibits the activity of executioner caspases. Also, EBER-mediated down-regulation of PKR counteracts both types of apoptosis pathways, as the PKR is an activator of caspase-8 that crosslinks them. The anti-apoptotic activities of EBV-encoded molecules, together with their ability to interfere with cell cycle regulators, may promote the outgrowth of immune and/or epithelial cells and the development of EBV-driven malignancies, such as BL, HL, DLBCL, LC, GC, NPC, NKTL, and AR-PCNSL. Therefore, they seem to be strong candidates for therapeutic targets for anti-viral and anti-tumor therapies.

## Figures and Tables

**Figure 1 ijms-23-07265-f001:**
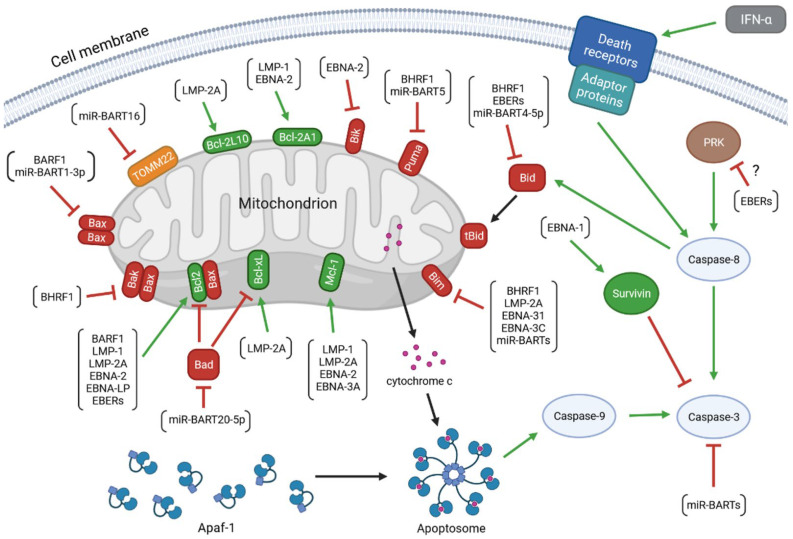
The anti-apoptotic activity of the EBV-encoded products. The colors of arrows represent the impact of the upstream molecules on the expression and/or activity of the target proteins (green: up-regulation; red: down-regulation). The molecular network determining the viability of the infected cell includes: (1) viral proteins: BamH1 fragment H rightward facing (BHRF1), BamH1-A reading frame-1 (BARF1), latent membrane proteins (LMP)-1 and -2A, EBV nuclear antigen (EBNA)-1, -2, -leader protein (LP), -3A and -3C; (2) viral non-coding transcripts: EBV-encoded small RNAs (EBERs), BamHI A rightward transcripts (miR-BARTs), and miR-BHRF1s; (3) anti-apoptotic members of the B-cell lymphoma 2 (Bcl-2) protein family: Bcl-2, B-cell lymphoma-extra large protein (Bcl-xL), myeloid leukemia cell differentiation protein 1 (Mcl-1), Bcl-2-like protein 10 (Bcl-2L10), and Bcl-2-related protein A1 (Bcl-2A1); (4) pro-apoptotic members of the Bcl-2 family: Bcl-2-associated X protein (Bax), Bcl-2 homologous antagonist/killer (Bak), Bcl-2-associated agonist of cell death (Bad), Bcl-2-like protein 11 (Bim), BH3-interacting domain death agonist (Bid), p53 up-regulated modulator of apoptosis (Puma), and Bcl-2 interacting killer (Bik); (5) other cellular molecules: mitochondrial import receptor subunit TOM22 homolog (TOMM22), apoptotic protease activating factor 1 (Apaf-1), cytochrome c, caspases(-8, -9, and -3), survivin, interferon (IFN)-α, death receptors, adaptor proteins, and dsRNA-dependent protein kinase (PKR).

**Figure 2 ijms-23-07265-f002:**
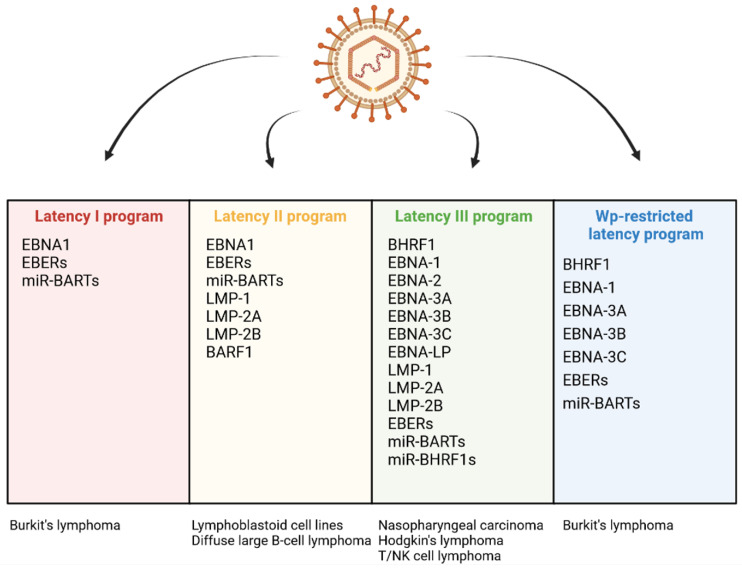
Differential expression programs in distinct types of EBV latency and various EBV-associated diseases.

## Data Availability

Not applicable.

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
