# Peer review of "Virus-Mediated Inhibition of Apoptosis in the Context of EBV-Associated Diseases: Molecular Mechanisms and Therapeutic Perspectives"

_ijms, 2022, doi:10.3390/ijms23137265_

Round 1
Reviewer 1 Report
This is very good review paper regarding EBV and its potential therapeutic targets in terms of its anti-apopotic proteins. I think it is acceptable with current version. I do not see any weak or revision points. Therefore I would accept this review with current context.
Author Response
Reviewer 1
This is very good review paper regarding EBV and its potential therapeutic targets in terms of its anti-apopotic proteins. I think it is acceptable with current version. I do not see any weak or revision points. Therefore I would accept this review with current context.
Response: We thank Reviewer 1 for appreciating our work and we are very grateful for the acquiescent review.
Reviewer 2 Report
Zbigniew Wyżewski et al. uncovered some interesting aspects of EBV biology.
Points to be addressed:
1) The rationale of why the authors came up with this review.
2) What is the information that is not exactly available that motivated the authors to come up with this information. What are the current caveats and how do the authors highlight the current research in answering them? If not they need to address in future directions.
3)This reviewer personally misses some aspects regarding In order to investigate the possible role of EBV in systemic lupus erythematosus (SLE) and its associated oral lesions, (please expand referring to PMID: 34768514);
4) The authors need to highlight what new information the review is providing to enhance the research in progress.
5) The underlying message here is that more precision and individualized approaches need to be tested in well designed clinical trials – a challenge, but I would be interested in their perspective of how this might be done.
6) I would suggest to slightly restructure the manuscript as follows:
P (Patient, population or problem)
Who or what is the patient, population or problem in question?
I (Intervention)
What is the intervention (action or treatment) being considered?
C (Comparison or control)
What other interventions should be considered?
O (Outcome or objective)
What is the desired or expected outcome or objective?
T (Time frame/treatment)
How long will it take to reach the desired outcome?
Author Response
Reviewer 2
Zbigniew Wyżewski et al. uncovered some interesting aspects of EBV biology.
Response: We thank Reviewer 2 for finding the topic of our article interesting and for providing constructive feedback. We have revised the manuscript text, the corrections are described here and are also highlighted in the manuscript in red color. We hope that changes made to the manuscript will meet the Reviewer’s expectations.
Points to be addressed:
1) The rationale of why the authors came up with this review.
Response: At the end of the “Introduction” (lines 54-62), we have added an explanation of our rationale for writing the article:
“The work has been written to systematize the current knowledge about the anti-apoptotic activity of EBV, synthesize different information spread in the scientific literature across multiple disciplines, and organize it into the form of an accessible review. It provides the consistent reconstruction of the molecular networks that determine the fate of EBV-infected cells. The number of facts described in our work provides the rationale for focusing on the development of anti-viral therapies that would antagonize the anti-apoptotic factors of EBV and lead to the elimination of infected cells within the human host. Thus, the article has been prepared with the intention to guide and inspire the new research on effective strategies against EBV infection.”
2) What is the information that is not exactly available that motivated the authors to come up with this information. What are the current caveats and how do the authors highlight the current research in answering them? If not they need to address in future directions.
Response: A number of facts described in our review are available in original articles, however, they are not a part of the wide-ranging, systematized descriptions of anti-apoptotic properties of EBV, and – as they have yet been not shown in extensive context – they seem to be less accessible for the reader. For example, the complete information that miR-BART1-3p binds to the 3’ UTR of DAB2 mRNA and causes the decrease in intracellular Bax is presented in no review article, therefore it is not blended into the wide panoramic description of anti-apoptotic properties of EBV. In our review, a set of previously reported facts has been situated in the new broad configuration.
3) This reviewer personally misses some aspects regarding In order to investigate the possible role of EBV in systemic lupus erythematosus (SLE) and its associated oral lesions, (please expand referring to PMID: 34768514);
Response: We have added the information about SLE and enriched bibliography with the position indicated by Reviewer (lines 85-87).
4) The authors need to highlight what new information the review is providing to enhance the research in progress.
Response: The review systematizes the knowledge about the set of EBV factors that counteract apoptosis of infected cells and contributes to the development of EBV-associated diseases. It provides the consistent reconstruction of the molecular network that includes the wide range of viral anti-apoptotic products and their cellular downstream proteins. Such a broad reconstruction has not been presented earlier. The article points to a wide range of potential therapeutic targets, setting the boundaries for possible future research on the treatment strategies based on the selective killing of the infected cells. As mentioned in the article, there are no advanced studies on the effectiveness of such therapies, however, the abundance of viral products combining both anti-apoptotic and pathogenic activities may encourage researchers to face this challenge in the future.
5) The underlying message here is that more precision and individualized approaches need to be tested in well designed clinical trials – a challenge, but I would be interested in their perspective of how this might be done.
Response: First, an extension of the range of in vitro experiments might be needed. The inhibitors of individual viral anti-apoptotic products, both alone and in different combinations, should be tested using various cell lines. The influence of the inhibitory molecule on the viability of infected cells (apoptosis rate) as well as on the virus yield, might be determined. The molecules displaying antiviral effects should be selected for clinical trials in order to eventually evaluate their effectiveness in the treatment of EBV-associated diseases.
6) I would suggest to slightly restructure the manuscript as follows:
P (Patient, population or problem)
Who or what is the patient, population or problem in question?
I (Intervention)
What is the intervention (action or treatment) being considered?
C (Comparison or control)
What other interventions should be considered?
O (Outcome or objective)
What is the desired or expected outcome or objective?
T (Time frame/treatment)
How long will it take to reach the desired outcome?
Response: Regrettably, the clinical trials have not been performed yet, and they must be preceded by extensive in vitro experiments to enable the selection of a possibly broadest set of therapeutic targets. We think that it is too early to determine the PICOT.
Round 2
Reviewer 2 Report
The authors have clarified several of the questions I raised in my previous review. Most of the major problems have been addressed by this revision.